# The Relationship between Gastrointestinal Health, Micronutrient Concentrations, and Autoimmunity: A Focus on the Thyroid

**DOI:** 10.3390/nu14173572

**Published:** 2022-08-30

**Authors:** Michael Ruscio, Gavin Guard, Gabriela Piedrahita, Christopher R. D’Adamo

**Affiliations:** 1Ruscio Institute for Functional Medicine, Austin, TX 94596, USA; 2Nova Institute for Health, Baltimore, MD 21231, USA; 3Department of Family & Community Medicine, University of Maryland School of Medicine, Baltimore, MD 21201, USA

**Keywords:** gastrointestinal health, hypothyroid, nutrients, IBS, nutrient–GI–thyroid axis

## Abstract

Currently, there is a lack of understanding of why many patients with thyroid dysfunction remain symptomatic despite being biochemically euthyroid. Gastrointestinal (GI) health is imperative for absorption of thyroid-specific nutrients as well as thyroid function directly. This comprehensive narrative review describes the impact of what the authors have conceptualized as the “nutrient–GI–thyroid axis”. Compelling evidence reveals how gastrointestinal health could be seen as the epicenter of thyroid-related care given that: (1) GI conditions can lower thyroid-specific nutrients; (2) GI care can improve status of thyroid-specific nutrients; (3) GI conditions are at least 45 times more common than hypothyroidism; (4) GI care can resolve symptoms thought to be from thyroid dysfunction; and (5) GI health can affect thyroid autoimmunity. A new appreciation for GI health could be the missing link to better nutrient status, thyroid status, and clinical care for those with thyroid dysfunction.

## 1. Introduction

The primary etiology of hypothyroidism is autoimmunity in Western populations where frank iodine insufficiency is not endemic. While many of these patients will require lifelong thyroid hormone replacement therapy, patients with subclinical hypothyroidism may improve their odds of not becoming frankly hypothyroid if nutritional therapy is administered.

Although many hypothyroid patients benefit from thyroid hormone replacement, a notable portion of hypothyroid patients (up to 40%) still struggle with symptoms despite being biochemically euthyroid when on thyroid replacement therapy [1,2,3,4,5]. As we will elucidate in further detail, evidence suggests that this may be secondary to interactions of what the authors have conceptualized as the “nutrient–gastrointestinal–thyroid” axis. 

Integrative and functional medical care has greatly focused on how micronutrients are related to thyroid function. However, clinical care often misses a key therapeutic target of improving thyroid health and nutrient status; that is gastrointestinal (GI) health. Moreover, evidence is currently lacking in how thyroid health, nutrient status, and gastrointestinal health are interconnected with one another. A narrative review of human studies and existing review papers was conducted to investigate these potential interconnections of the nutrient–GI–thyroid axis for consideration in clinical care and future research. 

## 2. Overview of Association between Thyroid Dysfunction, GI Dysfunction, and Nutrient Insufficiency 

There is a wealth of research highlighting the often-overlooked connection between thyroid function, GI health, and nutrient balance. 

An unexpected finding in 2017 concluded that being hypothyroid was the factor most highly associated with small intestinal bacterial overgrowth (SIBO), even more strongly associated than intestinal surgery or acid suppression medication use [6]. A year later, Polish researchers found that SIBO patients had a higher likelihood of thyroid autoimmunity [7]. A randomized controlled trial published in 2019 then showed hypothyroid patients with residual symptoms had reduced TSH, required a lower dose of thyroid replacement hormone (Levothyroxine), and improved fatigue scores after 2 months of probiotics supplementation [8]. The latter finding may be due to probiotics’ anti-inflammatory [9,10], anti-SIBO effects [11,12], and improved thyroid medication absorption [8]. 

Other GI imbalances are associated with autoimmunity, and thyroid autoimmunity in particular. For example, increased intestinal permeability markers are associated with a multitude of autoimmune conditions [13,14,15,16,17]. Specifically, intestinal permeability is found at higher rates in those with thyroid dysfunction and is associated with more thyroid symptoms [18]. In addition, a meta-analysis of 43 case-control studies showed that more virulent strains of *Helicobacter pylori* (*H. pylori*) can increase the risk of autoimmune disease including autoimmune gastritis and autoimmune thyroid disease [19]. 

This nutrient–thyroid–GI connection is also exemplified in a 2022 study suggesting 21% of autoimmune thyroid disease patients have anti-parietal cell antibodies (APCA) targeting the gastric mucosa [20]. As illustrated in Figure 1, the presence of APCA can predispose a patient to hypochlorhydria and subsequent key nutrient deficiencies, such as iron and B12. It has been shown that up to 4 in 10 hypothyroid patients are deficient in vitamin B12 which could contribute to “thyroid symptoms” such as fatigue [21]. 

An imbalance of one variable of the nutrient–GI–thyroid axis can lead to a downstream effect on the other parts of the axis. 

### 2.1. Important Nutrients Required for Thyroid Health 

A few select micronutrients, vitamins, and minerals have been proposed as essential for optimal thyroid function.

### 2.2. Iodine

Iodine is a non-metallic trace mineral that is an essential constituent of thyroid hormone synthesis and function. Iodine deficiency causes a variety of different disorders that include goiter and hypothyroidism. However, iodine is a double-edged sword for thyroid health; too much and too little are a problem. Iodized salt helps prevent goiter and other thyroid conditions. However, supplemental dietary iodine has also been shown to increase the incidence of thyroid autoimmunity [22,23,24,25].

In fact, a low-iodine diet can resolve hypothyroidism for a significant portion of people. In one study, over 75% of Hashimoto’s patients who were assigned a low-iodine diet became euthyroid in 3 months [26]. Another study showed that 53% of hypothyroid patients became euthyroid after just 3 weeks of iodine restriction. When iodine intake was resumed, they became hypothyroid once more [27].

Conversely, those following restrictive diets without adequate seafood or iodized salt intake run a risk of iodine insufficiency [28,29]. Thus, a proper balance of iodine is required for optimal thyroid function. 

### 2.3. Selenium

Selenium is a mineral with antioxidant properties that plays essential roles in the function of the immune system and in metabolism of thyroid hormone. Supplementation has shown to be helpful for both hypothyroidism and hyperthyroidism. A meta-analysis of 16 controlled trials showed that supplementation for 3 months reduced TPO antibodies (−271 IU/mL, 95% CI [−366, −175], *p*-value < 0.0001) for patients who were also being treated with thyroid replacement [30]. Other studies have corroborated these beneficial results [31,32]; however, not all data agree [33,34]. While selenium has been shown to reduce TPO antibodies, to our knowledge, it has not been robustly studied to show a reduction in progression of disease. 

On the other end of the thyroid spectrum, selenium reduced antibodies and the symptoms associated with Graves’ disease [35]. Moreover, those with the highest serum selenium levels were associated with Graves’ remission [36].

In addition, selenium may have a role in improving subclinical hypothyroidism and thyroid related symptoms such as depression [31,37,38]. These improvements may be maintained for months after just a short course of selenium supplementation [38]. 

### 2.4. Inositol 

Inositol is a sugar involved in cellular signaling. Subclinical hypothyroidism can often be treated synergistically with both selenium and inositol [31,37,39,40]. Using these agents together may be more effective than selenium alone [41]. Inositol may work by improving the thyroid gland’s sensitivity to TSH and lowering thyroid antibodies.

### 2.5. Vitamin D

Vitamin D is a secosteroid hormone that influences the expression of hundreds of genes. Low serum vitamin D levels are associated with hypothyroidism [42,43]. Treatment with supplemental vitamin D shows a large beneficial effect for reducing thyroid antibodies (both TPO and Tg antibodies) [44]. However, at least 3 months of treatment may be required to see a notable effect. 

### 2.6. Iron 

Iron is a trace mineral that has an essential role in human development, oxygenation of the blood, and production of hormones. Iron deficiency may be a key reason why some hypothyroid patients remain symptomatic. In fact, up to 50% of hypothyroid patients who are symptomatic despite thyroid replacement therapy have iron deficiency [45]. When hypothyroid women with persistent symptoms, despite appropriate levothyroxine therapy, increased their serum ferritin levels to greater than 100 mcg/L with iron supplementation, two-thirds of them had resolved fatigue [46].

It has been proposed that iron deficiency can significantly worsen thyroid function, including [47]: increase the risk of positive TPO antibodies, increased TSH, decreased fT4, and increased prevalence of overt and subclinical hypothyroidism.

Iron deficiency often has GI etiology as we will discuss later (e.g., gastritis, celiac disease). Thus, iron deficiency is a key variable in managing cases of thyroid dysfunction, especially when patients are symptomatic despite adequate thyroid replacement. 

### 2.7. Vitamin B12

Vitamin B12 is a water-soluble vitamin with critical roles in metabolism, methylation, and function of the nervous system. As stated earlier, up to 40% of hypothyroid patients have B12 deficiency [21]. Another study found that lower B12 and vitamin D levels were correlated with more TPO antibodies in a dose-dependent manner [48]. A third study showed that those with autoimmune thyroid disease had about 50% the B12 levels as healthy controls (200.70 vs. 393.41, *p*-value <0.0001) and was inversely correlated with TPO antibodies [49]. 

### 2.8. Zinc

Zinc is a mineral that is a component of hundreds of enzymes in the human body, exerting broad activity on the immune system and the production of a variety of neurotransmitters and hormones. While zinc has been proposed as a thyroid-supporting nutraceutical, some [50,51], but not all [52], studies show that zinc levels are associated with thyroid function. Thus, more research is needed to clarify supplemental zinc’s role in improving thyroid function. 

### 2.9. Magnesium

Magnesium is a mineral and cofactor for hundreds of enzymes that is involved in energy production, cell signaling, and DNA synthesis. Magnesium may play multiple roles in thyroid function including having an anti-inflammatory effect and reducing thyroid antibodies [53]. Magnesium deficiency is associated with higher thyroglobulin antibodies and hypothyroidism [54]. In addition, magnesium supplementation can improve symptoms attributed to thyroid dysfunction (e.g., fatigue, cognitive function, and constipation) [55,56,57]. 

Furthermore, one small case series found combined nutraceutical supplementation including magnesium improved thyroid function, reduced antibody titers, and led to normalization of thyroid ultrasound morphology in the majority of cases [58]. While this is merely one data point in a small cohort of patients, it suggests that a multiple-nutrient intervention could be used in clinical practice.

## 3. Impact of GI Conditions on Status of Micronutrients Important for Thyroid Health 

Many imbalances of the GI ecosystem have been found to directly impact the levels of the aforementioned key micronutrients important for thyroid health. 

### 3.1. Gastritis

Gastritis is inflammation of the gastric secondary to some form of gastric injury. It is often caused by an infection (e.g., *H. pylori*) or immune-mediated (autoimmune gastritis). Gastritis (either *H. pylori* or autoimmune) is associated with a deficiency of vitamin B12 and iron [59,60]. This can then lead to anemia and symptoms thought to be from thyroid dysfunction (i.e., fatigue, depressed mood). 

### 3.2. Irritable Bowel Syndrome (IBS) and SIBO

IBS is a functional GI disorder that is characterized by abdominal discomfort with associated changes in stool frequency and/or consistency. It is a multi-factorial process often involving microbial alterations, immune system changes, gut–brain dysfunction, mast cell reactivity, food sensitivities, altered motility, visceral hypersensitivity, and possibly inflammation. 

IBS and SIBO are associated with deficiencies of vitamin D [61,62], vitamin B12 [63], and serum zinc [64].

Furthermore, anxiety around foods that can trigger IBS symptoms can lead to avoidance of many nutrient-rich foods. This was exemplified in a recent cross-sectional analysis of 950 IBS patients that found that 13% exhibited severe food avoidance [65]. 

### 3.3. Inflammatory Bowel Disease (IBD)

IBD encompasses two distinct autoimmune conditions, ulcerative colitis (UC) and Crohn’s disease (CD). UC only affects the colon and inflammation is localized to the mucosal layer. CD can involve any part of the GI tract (anus to oral cavity) and involves transmural inflammation. IBD patients are at risk for malnourishment, increased catabolism, and reduced dietary intake which can compromise nutrient status [66]. Similar to IBS, those with IBD may exhibit food avoidance behavior. A systematic review of 19 studies found that active IBD patients consume less vitamin B12 as compared to healthy controls (less than 50% of recommended dietary intake) [67].

Furthermore, rates of zinc and selenium deficiency were more common in Crohn’s disease patients (selenium 15%, zinc 60%) compared to ulcerative colitis patients (selenium 6%, zinc 52%) and controls (selenium 0%, zinc 37%) [68]. This could be secondary to Crohn’s disease affecting the small intestine which is the main source of nutrient absorption.

### 3.4. Celiac Disease

Celiac disease is an autoimmune condition of the small bowel triggered by gluten consumption that can lead to villous atrophy, mucosal inflammation, and crypt hyperplasia. It can lead to atrophy of the small intestine microvilli and is associated with impaired iodine absorption [69]. In addition, up to 50% of celiac disease patients are iron deficient at the time of diagnosis [52].

### 3.5. Proton Pump Inhibitor (PPI) Use

PPI medications such as omeprazole and esomeprazole exert an effect by suppressing gastric acid secretion by inhibiting the H+/K+-ATPase in the gastric parietal cell of the stomach. Patients with iron deficiency taking PPIs can have suboptimal responses to iron supplementation [70]. PPI use is also associated with decreased zinc absorption [71] and lower serum B12 [72]. This may be secondary to an iatrogenic state of hypochlorhydria impairing the absorption of these key nutrients. 

### 3.6. Exocrine Pancreatic Insufficiency (EPI)

Finally, EPI is a condition whereby the pancreas produces a relative lack of enzymes necessary for proper digestive function, often a result of chronic pancreatitis. Those with EPI had a much higher rate of micronutrient deficiencies compared to health controls (42% EPI, 6% controls) [73]. The most common micronutrient deficiencies include those important for thyroid function such as selenium and magnesium. EPI may also predispose an individual to vitamin A and vitamin D deficiencies [74].

In summary, conditions affecting the GI system can result in frank and subclinical deficiencies of important nutrients important for thyroid function. 

## 4. GI Therapies Can Improve Nutrient Absorption

Fortunately, a number of minimally invasive GI therapies have been found to improve status of nutrients involved in thyroid function. 

### 4.1. Probiotics 

Probiotics are considered to be live microorganisms that may elicit a health benefit on the host. Probiotics may improve nutrient absorption and micronutrient status. For example, a systematic review of 14 studies found that intake of probiotics in healthy subjects was associated with a beneficial impact on micronutrient levels including vitamin B12, calcium, folate, iron, and zinc [75]. Another clinical trial showed improvement in plasma B12 and homocysteine levels after 40 days of probiotic-infused yogurt intake [76]. The same study also showed a marked decrease in anemia rates. 

The addition of synbiotics to iron supplementation led to greater iron levels [77]. This beneficial effect on iron status has been replicated in other studies [78,79]. Furthermore, probiotics have been shown to improve eradication of *H. pylori* [80,81] which is associated with gastritis and thus, impaired nutrient absorption. 

### 4.2. Elemental Diet

An elemental diet is a meal replacement formula of pre-digested protein and carbohydrates that is absorbed in the first few feet of the small intestine. It has shown a drastic reduction in global malnourishment rates. This was highlighted in a clinical trial of 144 Crohn’s disease patients who used a semi-elemental diet. After 4 months of therapy, the rate of those who were moderately or severely malnourished decreased from 91% to 24% [82]. An earlier 1995 trial showed an improvement in iron status after 4 weeks of elemental dieting in 19 patients with Crohn’s disease [83]. 

### 4.3. Immunoglobulins 

Serum-derived immunoglobulin/protein isolate (SBI) is a compound that binds to allergens and antigens, including bacterial toxins in the gut that renders them less able to irritate the gut mucosal lining and stimulate the immune system. The use of immunoglobulins in refractory IBS patients may improve nutrient absorption by reducing intestinal permeability [84,85]. Presumably, these benefits are secondary to improved dysbiosis and intestinal permeability, but more research is needed to clarify the exact mechanisms behind these findings. 

In summary, many key nutrients are vital for proper thyroid function (Figure 2). Multiple GI imbalances can lead to impaired absorption of these nutrients. Fortunately, GI care may lead to improved absorption of these nutrients which could indirectly affect thyroid health. 

## 5. Are Symptoms Emanating from the GI Tract or Thyroid?

The research provides compelling examples of how GI dysfunction causes thyroid-like symptoms. Seemingly challenging thyroid cases have been resolved by a GI-focused care model as noted in the research team’s previous published case series and literature review [86]. For example, a woman with long-standing hypothyroidism and multiple thyroid medication changes floundered in 15 months of thyroid-focused care. It was not until she started on a foundational GI protocol consisting of triple probiotic therapy, herbal antimicrobials, and intestinal repair nutrients that her symptoms resolved and her thyroid function (TSH, fT4, TPO antibodies) improved. 

The overlap of thyroid and GI symptoms is evident from the data. The five most common thyroid symptoms with the best predictive value of hypothyroidism include fatigue (81%), dry skin (63–76%), cold intolerance (64%), mood imbalance (46%), and hair loss (30%) [87,88]. 

However, these “hypothyroid symptoms” can also be from poor GI health. For example, IBS has been associated with fatigue [89,90], worse quality of life [91], depression [92,93], anxiety [94,95], and sleep disturbances [89,95,96]. SIBO has also been associated with both depression and anxiety [97].

In addition, other examples of GI imbalances leading to symptoms thought to be from thyroid dysfunction include the fact that non-celiac gluten sensitivity (NCGS) is associated with fatigue [98,99,100] alongside both anxiety and depression [98,99,101]. Moreover, malabsorption in IBD is associated with abnormalities of skin, hair, and nails [102,103].

The collection of above data supports the hypothesis that many symptoms attributed to thyroid dysfunction could in fact, be secondary to covert disturbances in the GI. 

## 6. GI Care Can Resolve Symptoms Thought to Be from Thyroid Dysfunction 

There are multiple lines of evidence that apparent hypothyroid symptoms (fatigue, depression, constipation) can be ameliorated with GI therapies.

Low FODMAP diet improves pain, anxiety, and quality of life [104,105,106]. Probiotics can improve depression and anxiety [105,107,108,109,110,111], cognitive function [112,113,114], SIBO eradication rate [7,8,9], and depression in those with SIBO [115]. Elemental diet improves quality of life [116,117]. Rifaximin improves cognitive function [118,119,120]. Improving intestinal permeability reduces chronic fatigue [121]. Fecal microbiota transplant improves fatigue and quality of life [122]. 

Multiple theories can explain the findings of GI therapies improving “thyroid symptoms”. These include: the symptoms were actually from GI imbalances, and not from thyroid dysfunction (as we will cover below); lower inflammation; modulation of the GI–brain axis; improvement of nutrient absorption; histamine intolerance which affects up to 43% of those with digestive disorders and can mimic hypothyroid symptoms [123]; and GI therapies improved absorption of thyroid medication.

It is clear that GI dysfunction can cause thyroid-like symptoms, and more importantly, GI treatments can resolve these same symptoms. We should also ask, “How likely is a clinician to see this presentation in their patients and what does the prevalence data suggest?”. Fortunately, there are compelling epidemiological data that address this question. 

## 7. GI Dysfunction Is at Least 45 Times More Common Than Hypothyroidism 

Frank hypothyroidism affects about 0.3% of the population [124,125,126,127] whereas IBS affects around 15% [128,129,130]. However, only about 30% of IBS cases are diagnosed [131], so IBS is likely more common than 15% of the population. It is important to note that some studies report much higher rates of hypothyroidism, but this is often reporting subclinical hypothyroidism which affects roughly 4% of the population [124,125,126,127]. Still, functional GI disorders such as IBS still affect a higher prevalence of the population. 

Therefore, a conservative estimate puts IBS as 45 times more common than frank hypothyroidism. Clinicians are thus considerably more likely to see cases of GI dysfunction than they are of overt hypothyroidism. In the absence of lab-confirmed frank hypothyroidism, the cause of symptoms such as depression and fatigue are more likely a result from GI imbalances than from thyroid dysfunction. The higher prevalence of GI dysfunction is further compounded by overdiagnosis of hypothyroidism, detailed below. 

## 8. Hypothyroidism Is Incorrectly Diagnosed and Overdiagnosed 

A growing trend in medical care is attempting to achieve “optimal” thyroid levels, sometimes through the means of prescribing thyroid replacement therapy. However, many hypothyroid patients are likely incorrectly diagnosed. As many as 61% of hypothyroid patients with a history of an ambiguous initial diagnosis can successfully discontinue medication and remain biochemically euthyroid with a majority reporting an improvement of their symptoms [132]. Another study found that two-thirds of patients started on levothyroxine had subclinical hypothyroidism where the mean TSH level at the initiation of treatment was 5.3 mIU/L [133]. 

This is important because many with subclinical hypothyroidism do not benefit from medication and have TSH normalize with time. This was highlighted in a study of 225 subclinical hypothyroid patients, where 74% had normal TSH at a 6-month follow up without any intervention [134]. A larger study of 422,000 people found that 62% of those with TSH between 5.5–10 had a normal TSH at a 5-year follow-up [135]. This is why the European Thyroid Association recommends a repeat TSH after 2–3 months in most cases of mild subclinical hypothyroidism [133]. This practice would likely lead to less individuals prematurely placed on unnecessary life-long thyroid medication. 

Moreover, there is a trend in the evidence that there is a lack of benefit for treating elderly individuals with subclinical hypothyroidism if TSH is below 7–10 mIU/L [136,137,138,139,140]. This may be because TSH naturally rises with age and could be considered as a normal anomaly that does necessitate treatment [140]. 

Third and finally, research has revealed that many patients do not need lifelong thyroid medication. Another meta-analysis found that 37% can successfully discontinue their medication and remain biochemically euthyroid even after a 5-year follow up [141]. This could mean that some individuals were likely placed on thyroid replacement medication either prematurely or inappropriately. The trend of incorrect and overdiagnosis of hypothyroid has led many clinicians and researchers to overlook GI health.

## 9. GI Therapies Improve Exogenous Thyroid Hormone Absorption 

Improving one’s GI health may be a pivotal part of improving thyroid medication absorption and should be considered before attempting to fine-tune thyroid type (i.e., desiccated, combo T4/T3) [142].

For example, probiotics have been shown to reduce TSH, reduce the required levothyroxine dose, and improve fatigue levels in hypothyroid individuals who were symptomatic despite appropriate Levothyroxine therapy [8]. Similarly, three studies have now shown that treating *H. pylori* improves TSH levels and may reduce required levothyroxine dose [143,144,145]. 

While there are no robust clinical trials on the SIBO-thyroid medication association, a recent case study sheds light on this powerful potential therapeutic target. A case study published in 2021 of a 51-year-old female with long standing Hashimoto’s and multiple food and medication sensitivities failed to achieve proper TSH balance over the course of 6 months, despite multiple medication changes (general T4, brand T4, combined T4/T3). It was not until she switched to a liquid T4 and then treated SIBO with rifaximin, that she achieved euthyroidism and resolved her symptoms [146]. Similar successful results were documented in the authors’ recent case series [86]. 

Clinicians may also find success by addressing potential food intolerances/sensitivities to improve thyroid medication malabsorption. Lactose intolerance is found in 76% of hypothyroid patients [147]. After 8 weeks of a lactose-free diet, a lower TSH was achieved in both euthyroid and subclinical hypothyroid participants. Similarly, a gluten-free diet leads to less necessary levothyroxine dose in those with atypical celiac disease [148]. 

Finally, proper output of hydrochloric acid is necessary for optimal thyroid replacement absorption. Studies have found that patients with impaired stomach acid secretion require higher doses of levothyroxine medication [144,145]. In a prospective study published in the New England Journal of Medicine, 10 patients with stable TSH started taking a PPI [145]. This led to a 37% increase in the median levothyroxine dose needed to maintain normal TSH levels. 

In light of these findings, it can be deduced that improving GI health (especially gastric and small bowel health) can improve thyroid medication absorption and thus, lower the need of medication to maintain euthyroidism in many patients. 

## 10. Impact of GI System on Autoimmunity and Thyroid Autoimmunity 

Imbalances of the GI ecosystem can directly and indirectly affect the immune system and are associated with states of autoimmunity, including thyroid autoimmunity [149]. As stated before, *H. pylori* is associated with multiple autoimmune conditions [19]. In addition, *Yersinia enterocolitica* was associated with a 4.3 times higher rate of thyroid autoimmunity [150], especially Graves’ (odds ratio 6.1, 95% CI [3.71–10.10], *p* < 0.0001). Other studies found that Graves’ and Hashimoto’s patients have higher rates of Yersinia IgG and IgA antibodies [151,152]. In addition, a systematic review found SIBO was present in 39% of patients with systemic sclerosis and associated with an average of 3.7 years longer disease duration [153].

The connection between GI health and autoimmunity may be mediated by increased intestinal permeability [13,14,15,16,17] caused by various forms of GI imbalances (i.e., SIBO, dysbiosis, pathogens). One study found that increased GI permeability is found at higher rates in those with thyroid dysfunction and is associated with more thyroid symptoms [18]. A small pilot study found that children with Hashimoto’s disease have increased markers of leaky GI when compared to controls [13]. In this study, higher serum zonulin was associated with higher levothyroxine dose. In other words, more intestinal permeability was associated with more thyroid dysfunction. 

Similarly, higher GI permeability levels among Graves’ patients were associated with higher antibody levels, lower TSH, higher fT4/fT3, and more symptoms [18]. This suggests that dysbiosis and intestinal permeability have a direct interaction and impact on thyroid autoimmunity (both Hashimoto’s and Graves’ disease) and can contribute to the autoimmune phenomenon as a whole. 

GI therapies directly reduce autoimmunity, suggesting that GI health may be a root cause behind the pathogenesis of autoimmunity. For example, a small study found an average of a 2000-point decrease in TPO antibodies after *H. pylori* eradication [154]. Probiotics have multiple lines of evidence showing that they lower inflammation [9,10] and autoimmunity [9,10,155,156]. An elemental diet was equal to oral prednisone in alleviating rheumatoid arthritis symptomatology [157]. More recently, a more invasive GI therapy of fecal microbiota transplant was found to reduce lupus antibody titers and improve disease activity [158]. These are just a few of the many studies showing an improvement in autoimmunity disease process and symptoms with a GI care model. 

In summary, optimizing GI health should be a main therapeutic target for those suffering from thyroid autoimmunity and autoimmunity more generally (Figure 3). 

## 11. Conclusions

This review has focused on the following points:-GI conditions can lower thyroid-specific nutrients.-GI care can improve status of thyroid-specific nutrients.-GI conditions are much more common than hypothyroidism.-GI care can resolve symptoms thought to be from “thyroid dysfunction”.-GI health can affect thyroid autoimmunity.

Specific nutrients are certainly imperative for proper thyroid health, but a sole focus on medication or nutrient intake as a way to improve thyroid function may miss a key therapeutic benefit of treating the patient’s GI health. Improving gastrointestinal health may indirectly improve thyroid-specific nutrient status and directly improve thyroid function, reduce autoimmunity, and symptoms thought to be from hypothyroidism (i.e., fatigue, brain fog). Given the prevalence of functional gastrointestinal disorders, it is likely that many hypothyroid patients are still floundering with chronic symptoms because they have yet to identify and treat the root cause of their symptoms; that is their GI health. 

The focus on gastrointestinal health in this review is not meant to be myopic. There are numerous environmental and genetic factors that also contribute to the etiology of thyroid disease that go beyond GI health (i.e., chronic stress, environmental toxicants, medications, and autoimmunity). Furthermore, more research is needed in this area of focus because normal thyroid function tests in serum may not necessarily indicate a euthyroid state in all peripheral tissues. For example, there is growing interest in the role of genetic polymorphisms in the deiodinase genes that would affect thyroid hormone concentrations in both blood and tissues [159]. Those with frank hypothyroidism, and some with subclinical hypothyroidism, will need lifelong thyroid medication. However, we contend that GI care cannot be overlooked and should be addressed alongside these other mediators of thyroid dysfunction. 

Clinicians may experience superior patient outcomes by understanding and implementing care that addresses the nutrient–GI–thyroid axis and researchers should consider evaluating the associations between these important domains in prospective studies.

## Figures and Tables

**Figure 1 nutrients-14-03572-f001:**
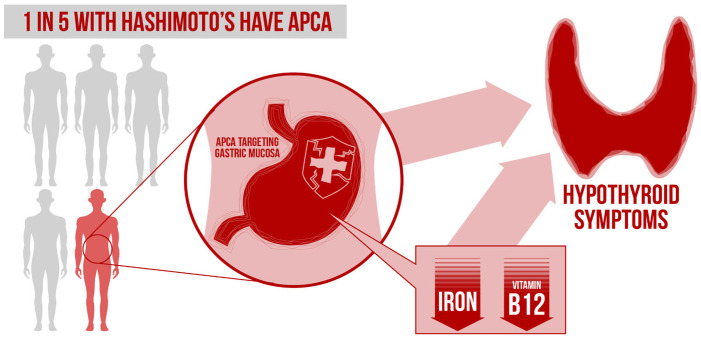
Relationship between Hashimoto’s, parietal cell antibodies, nutrient deficiencies, and apparent thyroid symptoms.

**Figure 2 nutrients-14-03572-f002:**
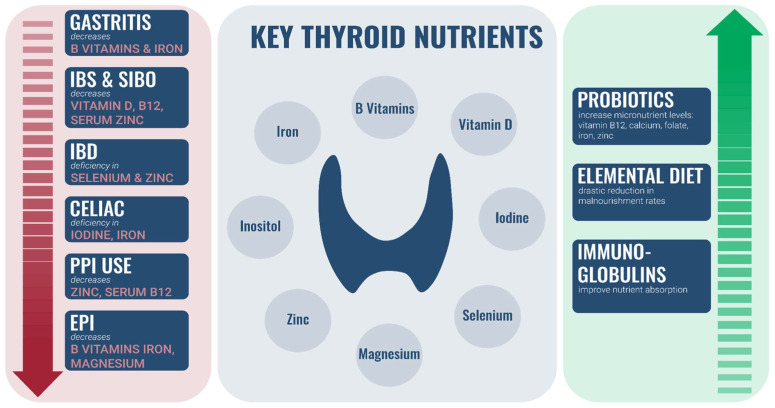
Relationship between nutrients, GI dysfunction, and GI-focused care.

**Figure 3 nutrients-14-03572-f003:**
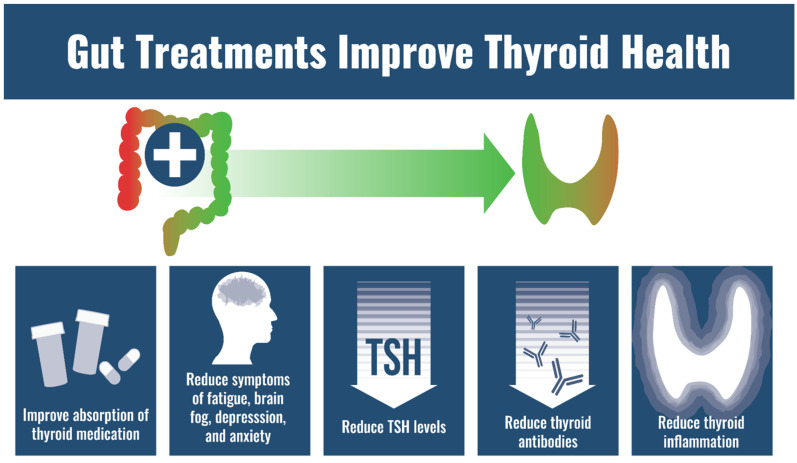
GI effects of GI care on thyroid health.

## Data Availability

Not applicable.

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
