# Peer review of "The Relationship between Gastrointestinal Health, Micronutrient Concentrations, and Autoimmunity: A Focus on the Thyroid"

_nutrients, 2022, doi:10.3390/nu14173572_

Round 1

Reviewer 1 Report

This is a great work, very timely and important. Excellent way to explore possible preventative measures in thyroid health and autoimmunity.

Here are a couple of comments that would make this manuscript even better:

1. First, even though it was clearly stated and defined as a narrative review, the manuscript does not provide any protocol, details or process description on how and why were certain articles reviewed and cited. It is expected under the Open Publication efforts that information published in any types of reviews should be available in proper details that allows replication and recreation of the article-base used to be reviewed and described. Please provide graphical or flow diagram chart format of the review process for transparency and replicability.

2. Each predictor variable, condition, chemical etc. used under the topic subheadings should be introduced clearly and with more details and appropriate scientific references. This would allow students, trainees, and residents to get more insights to each - otherwise well-selected - area of topic interests. It is critical to support a wide range of readers and audiences as the Journal regularly garners large international readership support. Medical and endocrinological education across the globe sadly vary, therefore reviews such as this by the Authors have an exceptional responsibility to educate and move the field forward equitably.

Author Response

The authors appreciate the opportunity to revise and resubmit the manuscript based on the thoughtful reviewer comments. We have provided point-by-point responses to each of the thoughtful comments and have tracked the changes in the revised manuscript to indicate where the corresponding changes were made. We believe that the manuscript has been substantially strengthened by the incorporation of these changes and are appreciative of the feedback. Please feel free to let us know if you would like additional information or clarification on any of the responses that follow.

This is a great work, very timely and important. Excellent way to explore possible preventative measures in thyroid health and autoimmunity.

Authors’ response: The authors appreciate the reviewer’s compliments on our manuscript. We are hopeful that the novel perspectives gleaned from this narrative review of the literature and offered in this paper generate further clinical research, and ultimately, improved root cause solutions for thyroid health and autoimmunity.

Here are a couple of comments that would make this manuscript even better:

  1. First, even though it was clearly stated and defined as a narrative review, the manuscript does not provide any protocol, details or process description on how and why were certain articles reviewed and cited. It is expected under the Open Publication efforts that information published in any types of reviews should be available in proper details that allows replication and recreation of the article-base used to be reviewed and described. Please provide graphical or flow diagram chart format of the review process for transparency and replicability.

Authors’ response: As the reviewer has noted, this was an explicitly stated narrative review. While the research team has extensive experience publishing systematic reviews and other types of literature reviews (PMID: 33849858, 32798042, 30893147, 28870406, 28188898, 28076926, 24473985), and adheres to the PRISMA guidelines offered by the EQUATOR network in our systematic reviews (https://www.equator-network.org/?post_type=eq_guidelines&eq_guidelines_study_design=systematic-reviews-and-meta-analyses&eq_guidelines_clinical_specialty=0&eq_guidelines_report_section=0&s=), no such guidelines currently exist for narrative reviews and flow chart formats for a review that spanned the intentionally broad and as-yet unconnected topic areas of nutritional, gastrointestinal, and thyroid literature is not possible.

However, the authors share the reviewer’s appreciation for offering as much transparency and replicability as is possible in a narrative review and have offered a clarifying statement in the Introduction section that the review was limited to human studies and existing review papers. The authors are hopeful that this additional clarity will offer greater reproducibility of this work going forward.

  1. Each predictor variable, condition, chemical etc. used under the topic subheadings should be introduced clearly and with more details and appropriate scientific references. This would allow students, trainees, and residents to get more insights to each - otherwise well-selected - area of topic interests. It is critical to support a wide range of readers and audiences as the Journal regularly garners large international readership support. Medical and endocrinological education across the globe sadly vary, therefore reviews such as this by the Authors have an exceptional responsibility to educate and move the field forward equitably.

Authors’ response: The authors completely agree that a brief introduction to the topic subheadings would be helpful for multidisciplinary readers who may be unfamiliar with any of the three main topic areas addressed in this paper within the nutrient-thyroid-GI axis. We are hopeful that having made these modifications will help support and educate readers in the topics that might be new to them.

Reviewer 2 Report

I share the mainstay of this manuscript but there are several issues to be clarify or correct.

- There are a lot of recent papers published in peer reviewed journal not cited

- There is a general confusion between different concepts: everyone knows that hypothyroidism symptoms are actually similar to those of other, often unrelated, disorders. Therefore, removing these generic symptoms by a nutritional approach does not imply an alternative treatment of hypothyroidism. This concept is explained in lines 230-231 but is better placed in the introduction: it improves clarity.

- There are several statement which are not adequately supported: i.e. the authors stated that up to 40% of treated hypothyroid patients have symptoms despite euthyroidism. This is a twenty years old paper dealing with psichological well being which is impaired in a similar percentage also in normal people and in fact the percentage of difference was significant but very small Moreover this was a questionnary where subjective feeling is predominant and the TSH was used as single test to measure  pharmacologic thyroid homeostasis, which may be measliding in a number of patients (polypharmacy, pregnant, patients with chronic disease, central hypothyroidism and dosage interference). Fitzgerald SP, et al.Thyroid testing paradigm switch from thyrotropin to thyroid hormones-Future directions and opportunities in clinical medicine and research. Endocrine. 2021 Nov;74(2):285-289. So this represent an unopposed opinion which does not fit with a balanced review.

- Most of the statement in the introduction are well known from more than 20 years (Carmel R, Spencer CA. Clinical and subclinical thyroid disorders associated with pernicious anemia. Arch Intern Med. 1982 Aug;142(8):1465-9). Also, it is not surprising that gastrointestinal disorders may be linked with altered thyroid homeostasis and autoimmunity. Worth to quote in the manuscript the following (Virili C, et al. Gastrointestinal Malabsorption of Thyroxine. Endocr Rev. 2019 Feb 1;40(1):118-136 and Virili C, et al. Gut microbiome and thyroid autoimmunity. Best Pract Res Clin Endocrinol Metab. 2021 May;35(3):101506).

- The role of selenium in Graves orbitopathy is well recognized, while its role in Hashimoto’s thyroiditis it is more controversial. It seems to be restricted to the reduction of anti TPO Abs, which however do not represent the progression of disease. Please add opposite results using selenium in hypothyroid patients for balance purposes.

- Line 106: Again, lowering circulating autoantibodies is not relevant to the progression of Hashimoto’s thyroiditis

- Iron deficiency and anemia causes fatigue but it is not directly derived from hypothyroidism. These patients have often GI disorders (gastric atrophy, celiac disease) which must be recognized independently from the presence of hypothyroidism. This must be clearly stated.

- Same as above for B12 deficiency ( Sibilla R, et al. Chronic unexplained anaemia in isolated autoimmune thyroid disease or associated with autoimmune related disorders. Clin Endocrinol (Oxf). 2008 Apr;68(4):640-5)

-line 135: same as above (line 106).  Magnesium is involved in muscle and bone metabolism and generic symptoms not necessarily can be attributed to thyroid.

Line 142: This has been recently reviewed, making clear that thyroid is a victim and not the killer (Virili C, et al. Levothyroxine Therapy in Gastric Malabsorptive Disorders. Front Endocrinol (Lausanne). 2021 Jan 28;11:621616.)

Line 151 and following: same as above (line 142)

Line 182: nothing to do with the thyroid. I would suggest to remove

Line 202: the whole point 4 is not related to the thyroid. Improving the nutrient absorption is of benefit for the whole body and thyroid is only one of them.

Ref. 147 cannot be quoted because in a review only published articles may be cited

Line 252: Subclinical hypothyroidism affects 5 to 10% of the population and represent the key point of confusion with symptoms coming from other organs, tissues or systems. Worth to mention

Point 8: levothyroxine replacement must be prescribed only to ascertained hypothyroid patients. Transitory hypothyroidism is a well known condition which ensue to drug use the presence of NSAT etc. I would suggest to reduce the para to two sentences including the last one which is perfect.

Point 9: excellent, but please quote the manuscript suggested in line 142

Point 10: excellent, but please quote the recent review suggested above: Virili C, et al. Gut microbiome and thyroid autoimmunity. Best Pract Res Clin Endocrinol Metab. 2021 May;35(3):101506).

Conclusions

I would suggest to remove the third one which is misleading

I share the mainstay of this manuscript but there are several issues to be clarify or correct.

- There are a lot of recent papers published in peer reviewed journal not cited

- There is a general confusion between different concepts: everyone knows that hypothyroidism symptoms are actually similar to those of other, often unrelated, disorders. Therefore, removing these generic symptoms by a nutritional approach does not imply an alternative treatment of hypothyroidism. This concept is explained in lines 230-231 but is better placed in the introduction: it improves clarity.

- There are several statement which are not adequately supported: i.e. the authors stated that up to 40% of treated hypothyroid patients have symptoms despite euthyroidism. This is a twenty years old paper dealing with psichological well being which is impaired in a similar percentage also in normal people and in fact the percentage of difference was significant but very small Moreover this was a questionnary where subjective feeling is predominant and the TSH was used as single test to measure  pharmacologic thyroid homeostasis, which may be measliding in a number of patients (polypharmacy, pregnant, patients with chronic disease, central hypothyroidism and dosage interference). Fitzgerald SP, et al.Thyroid testing paradigm switch from thyrotropin to thyroid hormones-Future directions and opportunities in clinical medicine and research. Endocrine. 2021 Nov;74(2):285-289. So this represent an unopposed opinion which does not fit with a balanced review.

- Most of the statement in the introduction are well known from more than 20 years (Carmel R, Spencer CA. Clinical and subclinical thyroid disorders associated with pernicious anemia. Arch Intern Med. 1982 Aug;142(8):1465-9). Also, it is not surprising that gastrointestinal disorders may be linked with altered thyroid homeostasis and autoimmunity. Worth to quote in the manuscript the following (Virili C, et al. Gastrointestinal Malabsorption of Thyroxine. Endocr Rev. 2019 Feb 1;40(1):118-136 and Virili C, et al. Gut microbiome and thyroid autoimmunity. Best Pract Res Clin Endocrinol Metab. 2021 May;35(3):101506).

- The role of selenium in Graves orbitopathy is well recognized, while its role in Hashimoto’s thyroiditis it is more controversial. It seems to be restricted to the reduction of anti TPO Abs, which however do not represent the progression of disease. Please add opposite results using selenium in hypothyroid patients for balance purposes.

- Line 106: Again, lowering circulating autoantibodies is not relevant to the progression of Hashimoto’s thyroiditis

- Iron deficiency and anemia causes fatigue but it is not directly derived from hypothyroidism. These patients have often GI disorders (gastric atrophy, celiac disease) which must be recognized independently from the presence of hypothyroidism. This must be clearly stated.

- Same as above for B12 deficiency ( Sibilla R, et al. Chronic unexplained anaemia in isolated autoimmune thyroid disease or associated with autoimmune related disorders. Clin Endocrinol (Oxf). 2008 Apr;68(4):640-5)

-line 135: same as above (line 106).  Magnesium is involved in muscle and bone metabolism and generic symptoms not necessarily can be attributed to thyroid.

Line 142: This has been recently reviewed, making clear that thyroid is a victim and not the killer (Virili C, et al. Levothyroxine Therapy in Gastric Malabsorptive Disorders. Front Endocrinol (Lausanne). 2021 Jan 28;11:621616.)

Line 151 and following: same as above (line 142)

Line 182: nothing to do with the thyroid. I would suggest to remove

Line 202: the whole point 4 is not related to the thyroid. Improving the nutrient absorption is of benefit for the whole body and thyroid is only one of them.

Ref. 147 cannot be quoted because in a review only published articles may be cited

Line 252: Subclinical hypothyroidism affects 5 to 10% of the population and represent the key point of confusion with symptoms coming from other organs, tissues or systems. Worth to mention

Point 8: levothyroxine replacement must be prescribed only to ascertained hypothyroid patients. Transitory hypothyroidism is a well known condition which ensue to drug use the presence of NSAT etc. I would suggest to reduce the para to two sentences including the last one which is perfect.

Point 9: excellent, but please quote the manuscript suggested in line 142

Point 10: excellent, but please quote the recent review suggested above: Virili C, et al. Gut microbiome and thyroid autoimmunity. Best Pract Res Clin Endocrinol Metab. 2021 May;35(3):101506).

Conclusions

I would suggest to remove the third one which is misleading

Risultati della trad

Author Response

The authors appreciate the opportunity to revise and resubmit the manuscript based on the thoughtful reviewer comments. We have provided point-by-point responses to each of the thoughtful comments and have tracked the changes in the revised manuscript to indicate where the corresponding changes were made. We believe that the manuscript has been substantially strengthened by the incorporation of these changes and are appreciative of the feedback. Please feel free to let us know if you would like additional information or clarification on any of the responses that follow.

  • There are a lot of recent papers published in peer reviewed journal not cited
    • Authors’ response: The authors appreciate the abundance of literature on this topic that we feel necessitates a review of this nature. As such, we have endeavored to be as inclusive as possible with respect to the literature we have cited. We appreciate the suggestions for additional citations that the reviewer has provided and have added them to the revised manuscript along with numerous others. We hope that these additions are satisfactory to the reviewer.
  • There is a general confusion between different concepts: everyone knows that hypothyroidism symptoms are actually similar to those of other, often unrelated, disorders. Therefore, removing these generic symptoms by a nutritional approach does not imply an alternative treatment of hypothyroidism. This concept is explained in lines 230-231 but is better placed in the introduction: it improves clarity.
    • Authors’ response: We appreciate and agree with the reviewer's perspective. Our clinical observation, based upon our patients' treatment history with other providers, is that many providers are not fully aware of this delineation, and thus prematurely pursue thyroid replacement therapy.  So while we agree, we feel this clarification will provide an important reminder for many clinicians. 
  • There are several statements which are not adequately supported: i.e. the authors stated that up to 40% of treated hypothyroid patients have symptoms despite euthyroidism. This is a twenty years old paper dealing with psychological well being which is impaired in a similar percentage also in normal people and in fact the percentage of difference was significant but very small. Moreover this was a questionnaire where subjective feeling is predominant and the TSH was used as a single test to measure pharmacologic thyroid homeostasis, which may be misleading in a number of patients (polypharmacy, pregnant, patients with chronic disease, central hypothyroidism and dosage interference). Fitzgerald SP, et al.Thyroid testing paradigm switch from thyrotropin to thyroid hormones-Future directions and opportunities in clinical medicine and research. Endocrine. 2021 Nov;74(2):285-289. So this represents an unopposed opinion which does not fit with a balanced review.
    • Authors’ response: This is a fair and important point. We have updated our language to clarify that data here are mixed, acknowledging that for many thyroid hormone replacement leads to significant improvement in symptoms and well-being, but that a subset still experiences symptoms despite achieving euthyroidism.  We have also included more current references to support our claim. Including;
      • https://www.ncbi.nlm.nih.gov/pmc/articles/PMC5824691/
      • https://pubmed.ncbi.nlm.nih.gov/29620972/
      • https://pubmed.ncbi.nlm.nih.gov/33237973/
      • https://www.ncbi.nlm.nih.gov/pmc/articles/PMC5965938/#!po=53.5714
        •  
      • Most of the statements in the introduction are well known from more than 20 years (Carmel R, Spencer CA. Clinical and subclinical thyroid disorders associated with pernicious anemia. Arch Intern Med. 1982 Aug;142(8):1465-9). Also, it is not surprising that gastrointestinal disorders may be linked with altered thyroid homeostasis and autoimmunity. Worth to quote in the manuscript the following (Virili C, et al. Gastrointestinal Malabsorption of Thyroxine. Endocr Rev. 2019 Feb 1;40(1):118-136 and Virili C, et al. Gut microbiome and thyroid autoimmunity. Best Pract Res Clin Endocrinol Metab. 2021 May;35(3):101506).
        • Authors’ response: The authors agree with this statement. These points in the introduction lay the foundation for which we layer on further evidence of the gut-thyroid-nutrients axis. We appreciate and have added these references.
      • The role of selenium in Graves orbitopathy is well recognized, while its role in Hashimoto’s thyroiditis it is more controversial. It seems to be restricted to the reduction of anti TPO Abs, which however do not represent the progression of disease. Please add opposite results using selenium in hypothyroid patients for balance purposes.
        • Authors’ response: We appreciate the reviewers' feedback here. We have added references demonstrating no change in TPO with selenium supplementation and correspondingly have acknowledged both sides of the data. This should add a fair balance to this controversial topic.
      • Line 106: Again, lowering circulating autoantibodies is not relevant to the progression of Hashimoto’s thyroiditis
        • Authors’ response: We appreciate the opportunity to clarify our position. To the authors' knowledge, there is a paucity of data looking into selenium supplementation's ability to reduce TPO and tracking correlation between the change in TPO to progression of disease.  However there are data showing that selenium supplementation can lower TPO and other separate data showing supplementation can lower TSH in SCH and help restore euthyroidism.  While the data here appear in their nascentsy, we feel it important to outline what trends in the data are finding, and draw a reasonable interference.  We have updated our language to more deliberately clarify the above and balance out our review of the literature. 
      • Iron deficiency and anemia causes fatigue but it is not directly derived from hypothyroidism. These patients have often GI disorders (gastric atrophy, celiac disease) which must be recognized independently from the presence of hypothyroidism. This must be clearly stated.
        • Authors’ response: We thank the reviewer again for helping us see where clarification is required. We agree that iron deficiency is not from hypothyroidism but rather may be one potential cause of low thyroid symptoms (fatigue) which may often have a GI etiology (such as APCA antibodies) and its prevalence increases in hypothyroid cohorts. We will add this clarification.
      • Same as above for B12 deficiency ( Sibilla R, et al. Chronic unexplained anemia in isolated autoimmune thyroid disease or associated with autoimmune related disorders. Clin Endocrinol (Oxf). 2008 Apr;68(4):640-5)
        • Authors’ response: We appreciate this feedback. We show evidence of GI imbalances causing B12 deficiency. For example, parietal cell antibodies (GI etiology) can lead to B vitamin insufficiency and lead to thyroid dysfunction and mimic thyroid symptoms. Furthermore, B vitamin deficiency has been noted at higher rates in hypothyroid cohorts. We hope this clarification will be sufficient to better explain our findings.
      • Line 135: same as above (line 106). Magnesium is involved in muscle and bone metabolism and generic symptoms not necessarily can be attributed to thyroid.
        • Authors’ response: We appreciate the perspective here.  We reference PMID: 26672672 as data supporting magnesium and its ability to aid in hypothyroidism.  We have clarified this is merely one data point and that this study used multiple nutrients as part of its intervention. . 
      • Line 142: This has been recently reviewed, making clear that thyroid is a victim and not the killer (Virili C, et al. Levothyroxine Therapy in Gastric Malabsorptive Disorders. Front Endocrinol (Lausanne). 2021 Jan 28;11:621616.)
      • Line 151 and following: same as above (line 142)
        • Authors’ response: These are great points and we fully agree. We later make clear how GI therapies improve thyroid medication absorption (Section 9). We have included this reference to and appreciate the reviewer’s suggestion here.
      • Line 182: nothing to do with the thyroid. I would suggest to remove
      • Line 202: the whole point 4 is not related to the thyroid. Improving the nutrient absorption is of benefit for the whole body and thyroid is only one of them.
        • Authors’ response: We appreciate the suggestion to condense our review and agree that the benefits of absorption are numerous. This section is imperative to our central conclusion; how the nutrient-gut-thyroid axis is reliant on all parts of the axis functioning optimally. One can improve the status of thyroid-important nutrients by focusing on rectifying gut health. Hopefully the findings and supporting references are sufficient to clarify how both a nutrient and gut-focused approach to thyroid care can act synergistically. A practicing clinician may not be aware of how GI-care can improve nutrient status and thus miss a key opportunity to improve thyroid-specific nutrient status.
      • 147 cannot be quoted because in a review only published articles may be cited
        • Authors’ response: This was just published and we have updated the citation.
      • Line 252: Subclinical hypothyroidism affects 5 to 10% of the population and represent the key point of confusion with symptoms coming from other organs, tissues or systems. Worth to mention.
        • Authors’ response: We appreciate the reviewers prompt for clarification. We have added clarification at the end of this paragraph showing a rough estimate of 4% prevalence of subclinical hypothyroidism with multiple supporting references from large observational studies (e.g. NHANES).
      • Point 8: levothyroxine replacement must be prescribed only to ascertained hypothyroid patients. Transitory hypothyroidism is a well known condition which ensue to drug use the presence of NSAT etc. I would suggest to reduce the para to two sentences including the last one which is perfect.
        • Authors’ response: We deeply appreciate the feedback as we feel this to be one of the most consequential points made in our paper.  Many clinicians are prescribing LT4 for subclinical hypothyroidism and even for euthyroidism that is not in the “optimal range”. This has been noted in two of the authors’ clinical experience as well as the supporting research noting that many are prescribed thyroid replacement with a mild elevation of TSH. The trend of the evidence is that there is a lack of benefit for treating subclinical hypothyroidism if TSH below 10 or in older subjects. We cite the following support in our paper.  We have also modified the language to better clarify this. 
      • Point 9: excellent, but please quote the manuscript suggested in line 142
        • Authors’ response: Thank you for this inspiring reference. We have quoted the paper accordingly. 
      • Point 10: excellent, but please quote the recent review suggested above: Virili C, et al. Gut microbiome and thyroid autoimmunity. Best Pract Res Clin Endocrinol Metab. 2021 May;35(3):101506).
        • Authors’ response: Thank you for this reference. We have quoted the paper accordingly. 
      • Conclusions;I would suggest to remove the third one which is misleading
        • Authors’ response: The authors have heeded the reviewer’s suggestion to remove the original version of the third conclusion. We have eliminated the “at least 45 times more common” portion of the conclusion to a more general and accommodating conclusion of “much more common”.
